# Sequencing and Chromosome-Scale Assembly of Plant Genomes, *Brassica rapa* as a Use Case

**DOI:** 10.3390/biology10080732

**Published:** 2021-07-30

**Authors:** Benjamin Istace, Caroline Belser, Cyril Falentin, Karine Labadie, Franz Boideau, Gwenaëlle Deniot, Loeiz Maillet, Corinne Cruaud, Laurie Bertrand, Anne-Marie Chèvre, Patrick Wincker, Mathieu Rousseau-Gueutin, Jean-Marc Aury

**Affiliations:** 1Génomique Métabolique, Genoscope, Institut François Jacob, CEA, CNRS, Univ Evry, Université Paris-Saclay, 2 Rue Gaston Crémieux, 91057 Evry, France; bistace@genoscope.cns.fr (B.I.); cbelser@genoscope.cns.fr (C.B.); lbertrand@genoscope.cns.fr (L.B.); pwincker@genoscope.cns.fr (P.W.); 2IGEPP, INRAE, Institut Agro, Université de Rennes, Domaine de la Motte, 35653 Le Rheu, France; cyril.falentin@inrae.fr (C.F.); franz.boideau@inrae.fr (F.B.); gwenaelle.deniot@inrae.fr (G.D.); loeiz.maillet@inrae.fr (L.M.); anne-marie.chevre@inrae.fr (A.-M.C.); mathieu.rousseau-gueutin@inrae.fr (M.R.-G.); 3Genoscope, Institut François Jacob, Commissariat à l’Energie Atomique (CEA), Université Paris-Saclay, 2 Rue Gaston Crémieux, 91057 Evry, France; klabadie@genoscope.cns.fr (K.L.); cruaud@genoscope.cns.fr (C.C.)

**Keywords:** genome, assembly, scaffolding, chromosome-scale, nanopore, optical map, bionano, omni-C, pore-C, plants

## Abstract

**Simple Summary:**

Reconstructing plant genomes is a difficult task due to their often large sizes, unusual ploidy, and large numbers of repeated elements. However, the field of sequencing is changing very rapidly, with new and improved methods released every year. The ultimate goal of this study is to provide readers with insights into techniques that currently exist for obtaining high-quality and chromosome-scale assemblies of plant genomes. In this work, we presented the advanced techniques already existing in the field and illustrated their application to reconstruct the genome of the yellow sarson, *Brassica rapa* cv. Z1.

**Abstract:**

With the rise of long-read sequencers and long-range technologies, delivering high-quality plant genome assemblies is no longer reserved to large consortia. Not only sequencing techniques, but also computer algorithms have reached a point where the reconstruction of assemblies at the chromosome scale is now feasible at the laboratory scale. Current technologies, in particular long-range technologies, are numerous, and selecting the most promising one for the genome of interest is crucial to obtain optimal results. In this study, we resequenced the genome of the yellow sarson, *Brassica rapa* cv. Z1, using the Oxford Nanopore PromethION sequencer and assembled the sequenced data using current assemblers. To reconstruct complete chromosomes, we used and compared three long-range scaffolding techniques, optical mapping, Omni-C, and Pore-C sequencing libraries, commercialized by Bionano Genomics, Dovetail Genomics, and Oxford Nanopore Technologies, respectively, or a combination of the three, in order to evaluate the capability of each technology.

## 1. Introduction

Assembling plant genomes has always been one of the most complex tasks in bioinformatics applied to genomics. Indeed, they often contain many repeated elements, such as satellites or transposable elements. This leads to an increase in the size of the genome, which can then reach tens of gigabases, for example the loblolly pine genome, which is 22 Gb in size [1]. Moreover, the difficulty is further increased due to the high levels of heterozygosity and highly variable ploidy [2]. All these characteristics make reconstruction of such genomes almost impossible without the help of a large consortium. However, the appearance of Illumina [3] technology fifteen years ago, combined with significant sequencing depth and the advent of assemblers using De Bruijn graphs, paved the way for low-cost genome assemblies. However, the resulting assemblies remained highly fragmented, and most repeats were not resolved.

Recently, the Pacific Biosciences (PacBio) and Oxford Nanopore Technologies’ (ONT) single-molecule sequencing technologies have been commercialized, which offer the opportunity to sequence fragments of several tens of kilobases, thus facilitating the assembly of complex genomes. However, this increase in the size of the sequenced fragments has a cost in terms of the read’s quality. Indeed, Nanopore and PacBio raw reads show an error rate of about 7% and 10%, respectively [4,5]. Due to errors remaining in long reads assemblies, a significant proportion of predicted genes may contain frameshifts [6], thus reducing the size of the predicted proteins, which in turn could cause problems in downstream analyses. To circumvent this issue, a number of polishing algorithms have surfaced. First, polishing algorithms were using the same type of data as was used to perform the assembly, as is the case in Nanopolish, for Nanopore data, or Quiver, for PacBio data. However, errors such as indels were still present in genome assemblies, and researchers quickly realized that pairing long reads data to generate the assembly and high-quality reads, such as the ones produced by Illumina sequencers, led to the least number of errors. This was, as an example, implemented in Pilon [7], Racon [8], or Hapo-G [9]. These gradual improvements in base quality and read length allowed researchers to generate high-quality plant genomes, reaching for the simplest (haploid genomes) chromosome scale without the need for long-range technologies [10,11].

Although the base quality of long reads has improved over the years, assembling and phasing heterozygous genomes still remains a difficult task. To try to solve this problem, PacBio drastically improved the quality of its generated sequencing reads by using a technique called Circular Consensus Sequencing (CCS) to generate high fidelity (HiFi) reads, thus breaking the 1% error rate barrier [12]. While standard PacBio reads are generated by a single pass of an enzyme around a circularized template, HiFi reads are produced by multiple passes of the enzyme through the same circularized sequences. This led to improvements in the base quality of the assemblies and made it possible to assemble and phase large genomes [13]. In a similar manner, ONT recently revealed their new Q20+ sequencing kit, which uses a new enzyme and optimized run conditions to lower the raw read error rate to 1%. It should be noted, however, that, at the moment of writing, we could not test the Q20+ kit and, therefore, could not validate the results shown by Oxford Nanopore.

However, this increase in read quality is not sufficient for all plant genomes. Indeed, highly heterozygous genomes or duplicated genomes suffer from regions that are often collapsed in genome assemblies. As an example, the recently sequenced tetraploid genome of potato [14] featured large regions (of the Mb order) that are identical between haplotypes, making the assembly impossible even with reads of several kilobases. To overcome this problem, a new technique called “gamete binning” has been recently developed. This technique, used to assemble the diploid apricot genome [15] and the autotetraploid potato genome [14], relies on the single cell sequencing of many gametes. This, in turn, makes it possible to reliably assign sequencing reads to a particular haplotype, and assemble long reads separately from the same haplotype. This simplifies the genome assembly process by reducing the complexity of the dataset, and enables a haplotype-by-haplotype assembly approach.

Although these technologies significantly simplify genome assembly, the resulting contigs often do not reflect the chromosomal organization of the original genome. Optical mapping is one of the long-range technologies now commonly used, in particular since the development of the Direct Label and Stain (DLS) protocol by Bionano Genomics (BNG). This protocol is designed to fluorescently label and repair high molecular weight (HMW) DNA at a specific location composed of six nucleotides (5′CTTAAG3′) using nicking endonucleases. In contrast to the previous version (Nick Label Repair and Stain, NLRS), this labelling preserves the double stranded DNA and avoids fragmentation of long DNA molecules [16]. These labelled molecules are charged into a flow cell and migrate into nanochannels in order to stay linear. The fluorescence signal is scanned, and the images are converted into molecule files (containing the position of the markers on each molecule) with a provided software. The optical map is generated by assembling the individual molecules into larger ones which can represent chromosomes. An optical card does not contain sequences, only labeling positions and distances between these positions, which can be seen as a barcode. The older NLRS protocol is used less and less as optical maps, generated thanks to this protocol, are generally less contiguous due to the labelling process that tends to induce breaks into DNA molecules. Indeed, the enzyme recognizes a double stranded sequence of seven nucleotides (5′GCTCTTC3′) and operates as a restriction enzyme. If several restriction sites are very close to each other, it could be a possible break point during the labelling step [17].

These two optical maps can be combined in the scaffolding process, allowing users to add the benefit of the two labelling methods. Sequences from a given assembly are digested in silico, and the estimated position of the labels on the contigs are compared to those of the optical maps. The scaffolding process detects assembly errors in the contigs, breaks them in order to be consistent with the optical maps, and finally orients and orders the contigs to produce chromosome or chromosome arm scale sequences.

The chromosomal conformation capture technique can also be used to obtain high-continuity assemblies and give information on chromosomal regions adjacent in the nucleus [18]. Indeed, chromosomes fold into topologically associating domains (TADs) [19], and then sequences form loops and other folds. The number of contacts decreases as a function of genomic distance. The chromatin is fixed in order to preserve its three-dimensional (3D) organization. An enzymatic digestion and a proximity ligation step produce a chimeric fragment containing two portions of DNA, distant on the chromosomal sequence, but close in the nucleus space. The relative abundance of each ligation product is related to the probability that those DNA sequences interact in the 3D space. After library preparation and short read sequencing, the paired reads represent long distance linking information. The mapping of the Illumina paired reads on the assembly highlights the long-distance links between two contigs or scaffolds. Library preparation kits are commercialized by several companies, for example Dovetail Genomics who propose a kit named Omni-C. It has the advantage of using an endonuclease for the digestion step instead of restriction enzymes, as in other kits, which avoids biases in digestion. It is interesting to note that optical maps and Omni-C provide different types of information, as the former contains linear information while the latter provides information about the spatial organization of the DNA molecules. However, algorithms dedicated to the scaffolding of genome assemblies do not take advantage of this kind of information, as they hypothesize that adjacent regions share a higher number of contacts than more distant regions.

The Pore-C library preparation is quite similar to the Hi-C library preparation in that it uses restriction enzymes, but the DNA fragments can be a concatenation of several chimeric junctions. After sequencing, it is possible, and expected, that Nanopore reads contain the multiple interacting sites. Indeed, as Nanopore reads are long, they span entire amplicons. To determine from which part of the genome the Nanopore reads originates, an in silico digestion of the reference genome is performed, and each segment of the read is assigned to a subsection of the reference. Although relatively young [20], the Pore-C technology seems to be interesting and competitive with traditional Hi-C for the scaffolding of complex genomes [21].

Traditionally, the anchoring of sequences along the chromosomes was performed using genetic maps [22]. These are obtained by genotyping a segregating population using genetic markers, such as SNPs (Single Nucleotide Polymorphism). The polymorph SNPS are then arranged on linkage groups according to the recombination frequency between markers. Thereafter, the localization of genetically mapped markers on scaffolds allows the validation, orientation, and anchoring of the scaffolds onto pseudomolecules [23]. However, some regions are still difficult to anchor due to a low density of genetic markers or the absence of recombination between markers, especially in (peri)centromeric regions [24] that are particularly rich in repetitive sequences [25], and thus require specific markers for anchoring [26]. 

Although sequenced with long reads in a previous study [27], the *Brassica rapa* (cv. Z1, AA, 2*n* = 20, double haploid line, genome size of 450 Mb) genome still contains a large number of unknown bases (33 Mb, representing 8.2% of the assembly). *Brassica* genus includes many important crops that are cultivated worldwide, notably for their oil production, or as vegetables. In addition, these species are one of the best models to study the importance of polyploidy in plant evolution, diversification, and adaptation, due to the occurrence of both ancient and recent polyploidization events [28]. Moreover, the *B.rapa* chromosomes [29] contain large centromeric regions that are notoriously difficult to assemble (at least, more complicated than the C genome, which contains, however, more transposable elements) [30], motivating us to resequence *B. rapa* cv.Z1 with the Oxford Nanopore technology and obtain the best assembly possible using current assemblers and long-range techniques. 

## 2. Materials and Methods

### 2.1. DNA Extraction for Nanopore Sequencing

High-quality and high-molecular-weight (HMW) DNA was extracted in order to generate long reads using ONT. For this purpose, DNA was isolated from one gram of plant leaves previously placed in the dark following the protocol provided by Oxford Nanopore Technologies (Oxford, UK), “High molecular weight gDNA extraction from plant leaves” downloaded from the ONT Community in March, 2019. This protocol involves a conventional CTAB extraction followed by a purification using the commercial Qiagen Genomic tip (QIAGEN, Germantown, MD, USA), and is described in detail in Belser et al. [11]. HMW gDNA quality was checked on a 2200 TapeStation automated electrophoresis system (Agilent, Santa Clara, CA, USA) and the length of the DNA molecules was estimated to be over 60 Kb.

### 2.2. Nanopore Sequencing

Two libraries were prepared simultaneously according to the following protocol and using the Oxford Nanopore SQK-LSK109 kit. Genomic DNA fragments (2 µg) were repaired and 3′-adenylated with the NEBNext FFPE DNA Repair Mix and the NEBNext^®^ Ultra™ II End Repair/dA-Tailing Module (New England Biolabs, Ipswich, MA, USA). Sequencing adapters provided by ONT (Oxford Nanopore Technologies Ltd., Oxford, UK) were then ligated using the NEBNext Quick Ligation Module (NEB). After purification with AMPure XP beads (Beckmann Coulter, Brea, CA, USA), the two libraries have been pooled into one. One third of the library was mixed with the Sequencing Buffer (ONT) and the Loading Bead (ONT) and loaded on a PromethION (Oxford Nanopore Technologies, Oxford, UK) R9.4.1 flow cell. A second third of the library was then loaded onto the flow cell after a Nuclease Flush using the Flow Cell Wash Kit EXP-WSH003 (ONT) according to the Oxford Nanopore protocol. The Nuclease Flush treatment consists of the use of a nuclease, which digests nucleic acids loaded on the flow cell. It allows the recovery of active pores, and increases the final yield of the run. ONT reads were basecalled using Guppy version 4.0.1 (Oxford Nanopore Technologies, Oxford, UK).

### 2.3. Nanopore Genome Assembly and Polishing

Three sets of reads were generated (Appendix A), in order to test assemblers with different read coverages and datasets. The first was composed of all reads, and for the second we selected a 30× coverage of the longest reads. The last was composed of 30× of the highest-scoring Filtlong [31] reads. Then, we launched Smartdenovo [32] (git commit 8488de9), Wtdbg2/Redbean [33] (git commit b77c565), and Flye [34] (version 2.8.3) on all sets. In addition, Necat [35] (git commit d377878) was launched with the complete readset, as it corrects reads given as input and applies its own downsampling algorithm. We launched Smartdenovo with “-k 17”, as advised by the developers in case of larger genomes, and “-c 1” to generate a consensus sequence. Redbean was launched with “-xont -X5000 -g 500m” and Flye with “-g 500m”. NECAT was launched with a genome size of 500 Mb, and other parameters were left as default.

The assembly produced by Necat was retained, as it was the closest to the expected size of the genome and the most contiguous (Appendix A). The Necat assembly was polished once by using Racon (version 1.4.13) [8] and Medaka [36] (version 1.2.0) with Nanopore reads and twice with 250 bp-long paired-end Illumina reads (PRJEB26620) by using Hapo-G (version 1.0) [9]. Racon was launched with the following parameters: “-m 8 -x -6 -g -8 -w 500 -u”, as advised by the Racon developers, and Medaka was launched with the “-m r941_prom_high_g360” parameter, in order to comply with the version of the basecaller that we used. Finally, Hapo-G was launched with the “-u” parameter.

### 2.4. Optical Mapping and Hybrid Scaffolding

Two optical maps were generated using previously generated molecules [27] and the assembly pipeline developed by Bionano Genomics (BNG) with the following two options: “add pre-assembly” and “non haplotype without extend and split”. The two-enzyme hybrid scaffolding pipeline (bionano solve and tools Version: 1.6.1) was used to generate the hybrid scaffolds (Table 1). BisCoT [37] was used to correct artifactual duplications (negative gaps) introduced during the scaffolding process. A final step of polishing was performed using Hapo-G and Illumina [27] sequencing data.

### 2.5. Omni-C Library Preparation and Illumina Sequencing

The Dovetail Omni-C library was prepared using the Dovetail Hi-C preparation kit (Dovetail Genomics, Scotts Valley, CA, USA), according to the manufacturer’s protocol (manual version 1.0 for non-mammalian samples), using young frozen leaves previously placed in the dark. Briefly, after sample crosslinking, chromatin was digested using a sequence-independent endonuclease. Proximity ligation (which creates chimeric molecules) was performed using a biotin-labeled bridge between the ends of the digested DNA. After reversal crosslinking, the DNA was purified and followed by library generation (omitting the fragmentation step). Finally, the biotinylated chimeric molecules were captured and amplified before sequencing on the Novaseq 6000 instrument (Illumina, San Diego, CA, USA) using 150 base-length read chemistry in paired-end mode.

### 2.6. Scaffolding Using the Omni-C Library

Scaffolding was realized thanks to the 3D de novo assembly (3D-DNA [38]) pipeline (version 180419). Hi-C raw reads were aligned against the assembly (-s none option) using Juicer. The resulting merged_nodups.txt file and the assembly were given to the run-asm-pipeline.sh script with the options “--editor-repeat-coverage 5 --splitter-coarse-stringency 30 --editor-coarse-resolution 100,000”. Contact maps were visualized through the Juicebox tool [39] (version 1.11.08) and edited to adjust the construction of scaffolds or break misjoins. After edition, the new.assembly file was downloaded from the Juicebox interface. The file is filtered and converted into a fasta file thanks to the juicebox_assembly_converter.py script [40].

### 2.7. Pore-C Library Preparation and Nanopore Sequencing

The RE-Pore-C library was carried out as described in the Oxford Nanopore Technologies RE-Pore-C protocol for plant samples (30 July 2020 version), with the exception of tissue fixation, which was performed following the protocol of Chang Liu [41]. Crosslinked plant nuclei were isolated prior to in situ restriction digestion with NlaIII (New England Biolabs, Ipswich, MA, USA). After overnight incubation, the restriction enzyme was heat-denatured; crosslinked DNA clusters were ligated in proximity, followed by protein degradation and de-crosslinking, releasing the chimeric Pore-C dsDNA polymers. Libraries were constructed using the ONT Ligation Sequencing Kit (SQK-LSK109), following the library preparation recommendations. Sequencing was carried out on a R.9.4.1 PromethION flowcell. Nuclease washes were used to maximize output.

### 2.8. Scaffolding Using the Pore-C Library

Prior to scaffolding, we removed all nanopore Pore-C reads that were larger than 100 kb in size, as it drastically increases running times and can be problematic for the pipeline to handle, as stated in a github issue [42]. Then, we used the Pore-C Snakemake [43] pipeline (git commit 6b2f762) developed by ONT to generate the necessary files for scaffolding with the Salsa2 [44] scaffolder. Both Salsa2 and the Pore-C Snakemake pipeline were launched with default parameters.

### 2.9. Super-Scaffolding

In addition to testing the scaffolding of the nanopore assembly with one long-range technology, we combined several techniques to test if combining scaffolding methods would lead to better results. In particular, we super-scaffolded the BNG scaffolds with the Pore-C and Omni-C libraries and vice versa. Tools and parameters were the same as already described in previous sections. The 3D-DNA additional option -r 0 was used for scaffolding the BNG scaffolds with the Omni-C library.

### 2.10. Validation of the Assemblies

#### 2.10.1. Comparison to Reference Genomes

We used the *B.rapa* cv. Chiifu v3 [45] and the *B.napus* cv. Darmor-BZH v10 [30] A-subgenome to generate dotplots and compare our different assemblies to established reference genomes. To do so, we used minimap2 with the “-x asm20” parameter to generate alignments of the assemblies against each reference. Then, we used D-Genies [46] in order to visualize the alignments and enabled the “Sort contigs” and “Hide noise” options.

#### 2.10.2. Quality Assessment with Merqury

We used merqury (version 1.3, git commit 6b5405e) to obtain a quality score for each of our assemblies. First, we used the bundled best_k.sh script with a tolerable collision rate of 0.0001 and a genome size of 500 Mb to find the best size of kmer for our genome, which gave us a kmer size of 21. Then, we used meryl (version 1.3, git commit 3400615) to compute the 21-mer counts with Illumina reads via the meryl count command with default parameters. Finally, we used merqury to compare the kmers of each assembly to the kmers extracted from the Illumina reads.

#### 2.10.3. Gene-Completeness Estimation with Busco

In order to estimate the gene completeness of the genome assemblies, we launched Busco version 5.1.2 with the embryophyta datasets (odb10). All other options were left as default.

### 2.11. Construction of a B. rapa Genetic Map and Anchoring

To construct a *B. rapa* genetic map, we created a F2 population (149 plants) deriving from an initial cross between the doubled haploid *B. rapa ssp trilocularis* cv. Z1 and the inbred line *B. rapa ssp pekinensis* cv. Chiifu-401-42. DNA from the parental lines, the F1 hybrid and the 149 plants were extracted using the sbeadex plant kit (LGC Genomics, Teddington Middlesex, UK) on the oKtopure robot at the GENTYANE platform (INRAE, Clermont-Ferrand, France), and thereafter genotyped using the *Brassica* 19 K Illumina infinium SNP array (TraitGenetics, Gatersleben, Germany). From these data, a total of 4030 markers were found polymorph between the parental lines, and were used to create a genetic map (985.9 cM in total) using CarthaGene [47] software version 1.2.3, with a LOD score of 4 and a maximal genetic distance of 0.21 cM. For all these genetically mapped markers, we then blasted their sequence contexts against the *B. rapa* cv. Z1 scaffolds obtained using either additional Bionano data only or both Bionano and Omni-C data, totaling 3867 and 3865 physically anchored markers, respectively. This step allowed us to help with the ordering and orientation of scaffolds, as well as to compare the quality of both assemblies.

### 2.12. Putative Position of Peri-Centromere and Sub-Telomere Regions

The putative position of (peri)centromeres was inferred by blasting the centromere-specific repeat sequences CentBr1 and CentBr2 (CW978699 and CW978837, respectively [48]) against this novel assembly, as well as sequences that are found in the pericentromeric heterochromatin blocks of *Brassica* chromosomes: the centromere-specific Ty1/copia-like retrotransposon of *Brassica* (CRB, AC166739), the pericentromeric Ty3/Gypsy-like retrotransposon of *B.rapa* (PCRBr, ACC166740), a 238-bp degenerate tandem repeat (TR238, AC166740), and a 805-bp tandem repeat (TR805, AC166739 [49]) (Lim et al. 2007). The putative positions of subtelomeres were inferred by blasting the *B. rapa* subtelomeric satellite repeats (pBrSTRa/b, EU294384 and EU294385 [50]) against this novel assembly.

### 2.13. Gene Prediction

Gene prediction was performed using several proteomes: 8 from other genotypes of *B. napus* [51] (Westar, Zs11, QuintaA, Zheyou73, N02127, GanganF73, Tapidor3, and Shengli3), *Arabidopsis thaliana* (UP000006548), and the 2021 annotation of Darmor-bzh [30]. Regions of low complexity in genomic sequences were masked with the DustMasker algorithms [52] (version 1.0.0 from the blast 2.10.0 package). Proteomes were then aligned on the genome in a 2-step strategy. First, BLAT [53] (version 36 with default parameter) was used to quickly localize corresponding putative regions of these proteins on the genome. The best match, and the matches with a score ≥90% of the best match score, have been retained. Second, alignments were refined using Genewise [54] (version 2.2.0 default parameters), which is more accurate for detecting intron/exon boundaries. Alignments were kept if >75% of the length of the protein was aligned on the genome.

All the protein alignments were combined using Gmove [55], which is an easy-to-use predictor with no need for a pre-calibration step. Briefly, putative exons and introns extracted from alignments were used to build a graph, where nodes and edges represent exons and introns, respectively. Gmove extracts all paths from the graph, and searches open reading frames that are consistent with the protein evidence. Finally, we decided to exclude single-exon genes composed of >80% of untranslated regions. Following this pipeline, we predicted 56,073 genes with 4.39 exons per gene on average.

## 3. Results

### 3.1. Nanopore Sequencing and Long Reads Assembly

A single PromethION R9.4.1 flowcell produced 93 Gb of data with an N50 of 26.9 kb (Appendix A), representing a genome coverage of approximately 186×, with 38× being composed of reads longer than 50 Kb.

We subsampled data as previously described (30× longest, 30× highest scoring Filtlong reads, and all reads). Flye and Redbean produced their most contiguous assembly with a subset of reads, showing that experimenting with different coverages may be beneficial for the assembly. Necat with the entire readset led to the most contiguous assembly, with a cumulative size of 442 Mb, a contig N50 of 10.4 Mb, a merqury quality score of 27.5, and 1567 (97.1%) complete and single-copy embryophyta BUSCO genes (Appendix A). After polishing with Nanopore and Illumina reads (Appendix A), the merqury quality score rose to 36.4 and the number of complete embryophyta BUSCO genes increased to 1604 (99.4%). By aligning the Necat assembly to the *B. rapa* cv. Chiifu v3 and the *B. napus* Darmor-bzh v10 reference genomes (Appendix A), we could detect the presence of a contig, showing a translocation between the A07 and A03 chromosomes. We concluded that this contig was chimeric as it was detected as such and cut later by the Hi-C and BNG software. However, this chimeric junction was left as is, and used to see if scaffolding algorithms would be able to correct it.

### 3.2. Long Range Genome Assembly

#### 3.2.1. Hybrid Scaffolding

The produced DLE and BspQI maps achieved a N50 of 13.5 Mb and 1.9 Mb and a cumulative size of 466 Mb and 434 Mb, respectively. The combination of the two optical maps with the nanopore contigs lead to an assembly that reached a N50 of 16.8 Mb with a cumulative size of 446.4 Mb (Appendix A). Very few gaps were introduced underlining the great concordance between the length of the optical maps and the nanopore assembly. Finally, as expected, BisCoT software increased the N50 and decreased the cumulative size. Indeed, this was expected, as BisCoT is designed to remove artifactual tandem duplications (negative gaps) and the possible redundancies in the assembly. A final round of polishing gave an assembly with a N50 of 17 Mb and a cumulative size of 443.9 Mb (Appendix A). The comparisons with the *Chiifu* genome and Darmor-bzh A-subgenome showed neither a chimeric scaffold nor a misorganization (Appendix A).

#### 3.2.2. Hi-C Scaffolding

The sequencing of the Omni-C library produced 108 M paired reads (32 Gb) that were aligned on the nanopore polished contigs using the Juicer pipeline. The mapping information was used by 3D-DNA [38] to orient and order sequences of the input assembly. A contact map was generated and visualized through the Juicebox [39] interface. We edited the contact map and merged some scaffolds (two larger scaffolds were constructed by merging two scaffolds for each) and broke one misjoin (Appendix A). We obtained an assembly of 439 Mb with a N50 of 25.5 Mb (Table 1). Comparisons with the *Chiifu* genome and *B. napus* A-subgenome showed neither a misjoin nor a misorganization (Appendix A).

#### 3.2.3. Pore-C Scaffolding

The sequencing of the Pore-C library (Appendix A) produced 15.55 Gb of data with an N50 of 3.99 Kb. As advised by the developers of the Pore-C Snakemake pipeline, we removed reads that were longer than 100 Kb and obtained a resulting dataset composed of 5.8 million reads with an N50 of 3.97 Kb, for a total cumulative size of 15.50 Gb. This readset was used to perform all scaffoldings with the Pore-C technology presented in this study.

We applied the Pore-C Snakemake pipeline released by Oxford Nanopore, as well as Salsa2 to scaffold the Necat assembly (Table 1), and obtained 253 scaffolds for a total cumulative size of 443 Mb, with a scaffold N50 of 20.1 Mb. After aligning scaffolds onto the reference genomes, and inspecting alignments and the resulting dotplots (Appendix A), we could not find any evidence indicating the presence of chimeric sequences.

### 3.3. Combination of Several Long-Range Techniques

Additionally, we investigated whether the combination of long-range technologies could add the benefits of each (Figure 1). For this purpose, we scaffolded the Omni-C and Pore-C assemblies with the optical maps. Hi-C scaffolding sometimes introduces misjoins (chimera or chromosome arm inversion) that could be corrected by the comparison with optical maps. Surprisingly, the scaffolding of the Omni-C assembly (Omni-C + BNG assembly) decreased the contiguity and added more than 2% of undetermined bases. It produced an assembly of 445 Mb with a N50 of 20.3 Mb (Appendix A). On the other hand, the Pore-C + BNG assembly produced an assembly of 450 Mb and a N50 of 25.9 Mb (Appendix A). The N50 slightly increased, but scaffolds did not achieve the chromosome scale. When comparing final assemblies to reference genomes, no misjoins were detected, with the exception of a part of the A01 chromosome being duplicated in the Pore-C + BNG assembly. Conversely, we scaffolded the BNG scaffolds with the Hi-C or the Pore-C libraries (Table 2). As the BNG scaffolds were close to chromosome scale, we expected that, with the long distance contacts of the Hi-C, we could organize and orient the BNG scaffolds into complete chromosomes. We obtained a BNG + Omni-C assembly of 440 Mb with a N50 of 33.3 Mb after editing the contact map (Appendix A). The size of the 10 largest scaffolds were compatible with chromosome length, and dotplots showed good consistency with the *B. napus* A-subgenome and the *Chiifu* genome (Appendix A). Likewise, we obtained a BNG + Pore-C assembly of nearly 444 Mb with a N50 of 17 Mb (Appendix A). Again, the alignments with reference genomes did not show any inconsistencies, but no scaffold achieved the chromosome scale. As metrics were similar to those obtained with the BNG scaffolds, we put aside the BNG + Pore-C assembly.

### 3.4. Anchoring

Based on the previous results, the anchoring with the genetic map was performed on two assemblies: The BNG scaffolds and the BNG + Omni-C scaffolds. The comparison of the position of the markers allows us to verify the structure of the scaffolds, to assign and orientate scaffolds onto chromosomes. We were able to anchor 384 Mb of the BNG scaffolds and 369 Mb of the BNG + Omni-C scaffolds on the chromosomes (compared to 357 Mb of anchored sequences in the V1 version). The putative chromosomes are composed of two to five BNG scaffolds or one to two BNG + Omni-C scaffolds. No misjoins were detected in the BNG scaffolds, but one inversion was detected in the BNG + Omni-C scaffolds. To construct the final version, we decided to keep the 36 anchored BNG scaffolds that contained more sequences. However, we used the Omni-C data to add two scaffolds in chromosome A03 and one scaffold in chromosome A09 which were difficult to anchor with the genetic map. In the end, we succeeded to anchor 386.05 Mb (86.96 % of the whole genome assembly). The final *B. rapa* cv. Z1 version 2 is composed of 210 scaffolds with a cumulative size of around 444 Mb and a N50 of 39,2 Mb (L50 = 5) (Table 3 and Figure 2). Among the ten chromosomes, seven chromosomes contained telomeric repeats (TTTAGGG motif) at both ends, and the remaining three had telomeric repeats at one end. In the same way, subtelomeric repeats have been found in each chromosome (except for chromosome A03) (Figure 2). Compared to the *B. rapa* cv. Z1 version 1, the cumulative size and the contig N50 have greatly increased (6.6 Mb and 10.2 Mb for versions 1 and 2, respectively) showing that the new assembly covered a higher proportion of the estimated genome length (Table 3). In the same way, the rate of undetermined bases dropped from 8.22 to 0.66%. The final comparison of the genetic and physical positions revealed that only 179 out of the 3867 markers (4.63%) were discordant, most often due to an inaccurate position on the genetic map (of a few cM).

## 4. Discussion

In this study, we used the genome of *B. rapa* cv. Z1 to feature how plant genomes can now be generated. We chose the Oxford Nanopore PromethION technology and also compared current long-range scaffolding techniques, namely Bionano optical mapping and the Hi-C and Pore-C chromatin conformation capture techniques, in order to obtain chromosome-scale assembly.

We sequenced the genome of *B. rapa* cv. Z1 on a single R9.4.1 PromethION flowcell and showed that sequencing a medium-sized genome is now affordable to individual laboratories as the total cost of sequencing, including consumables, was about USD 1110. The great amount of data we obtained, added to the size of the reads, made it possible for us to assemble the genome using Necat into large contigs, with an N50 of 10.4 Mb. However, after aligning this assembly to the *B. rapa* cv. Chiifu or the *B. napus* cv. Darmor-bzh A-subgenome, we detected a chimeric contig, showing a translocation from the A07 to the A03 chromosome. We polished the contigs using a combination of Nanopore data and Illumina data, and reached a quality score of 36.4, representing an error rate of 0.02%. Remarkably, the quality score was already high, even when polishing only with Nanopore reads (31.04), showing that the Nanopore sequencing technology has made sizable progress since its beginnings [56].

The polished contigs were then scaffolded using optical maps, or two chromatin conformation capture techniques, Omni-C and Pore-C. Regarding the continuity of the resulting scaffolds, the ones obtained with Omni-C had the largest N50 (25 Mb), but the BNG and Pore-C scaffolds were very close to it, with an almost identical L50. However, a scaffold N90 of 3.4 Mb seemed to indicate that scaffolding with optical maps makes it possible to anchor a higher proportion of small contigs. In comparison, the N90 of the Pore-C and Omni-C assemblies were 1.47 Mb and 222 kb, respectively. Scaffolding with optical maps also introduced 2.9 Mb of unknown bases (Ns), as it is the only technique tested here that can estimate gap sizes, with other techniques only inserting an arbitrary amount in each gap, giving the false impression that the assembly is complete. Finally, when scaffolds were compared to reference genomes, we were pleased to see that every technique was able to successfully correct the chimeric region that was initially present in Nanopore contigs.

As we wanted to see if combining long-range technologies would lead to a better assembly, we scaffolded polished Nanopore contigs by combining two long-range techniques. We were able to reach chromosome-scale, first by using optical maps, and then Omni-C. Indeed, we obtained a single scaffold per chromosome assembly (N50 of 33.3 Mb and L50 of 5), with 2.9 Mb (0.67%) of undetermined bases.

In contrast, we noticed that Omni-C or Pore-C scaffolds were fragmented when integrated with optical maps (Appendix A). Indeed, the sequences are broken when conflicts are detected between the optical maps and the assembly. As we know that, in our case, the Hi-C scaffolding did not produce chimeras (Appendix A), we assume that the conflicts may result from a poor estimate of the gap size during the Hi-C scaffolding process. In this case, the scaffolds are chained with fixed gap sizes (100 nucleotides) even if some small contigs can take place in these gaps, with the consequence of generating conflicts with the optical maps. This is as BNG software cannot handle underestimated gap sizes. Therefore, we do not recommend chaining Hi-C scaffolding and then optical maps in this order. Conversely, Hi-C libraries generally contain more distant information than optical maps, and the integration of Hi-C data on the BNG assembly did not suffer from incompatibility issues. Likewise, this combination can also be used to validate the first scaffolding, as BNG scaffolds can already reach the chromosome scale.

The use of the genetic map allowed us to obtain complete chromosomes from the BNG assembly, and to validate the BNG + Omni-C scaffolds. These assemblies were chosen for their global metrics: BNG scaffolds had good N50 and N90, BNG + Omni-C scaffolds already reached chromosome scale. The quantity of anchored bases is lower with the BNG + Omni-C assembly, showing that the Omni-C scaffolding had fragmented scaffolds during the scaffolding process. Highly repeated regions are more likely difficult to organize with Hi-C data, and the very large centromeres in Z1 were problematic. As an advantage of the optical map, complex regions as centromere or rDNA genes clusters are generally well covered, thanks to the very long DNA molecules (>200 Kb long), allowing the organization of long-read assemblies.

In our opinion, optical mapping is the recommended long-range technology that leads to complete assemblies, mostly in complex regions such as centromeres, but requires additional tools which are not provided by BNG. Indeed, bioinformatic tools dedicated to optical maps are rare, which makes the use of this data more complicated than Hi-C short-reads. Additionally, it could be difficult to extract high molecular weight DNA in particular for plant genomes. Obtaining very long DNA molecules without any contaminant as polyphenol compounds that can inhibit the enzymatic reactions is critical. In our own hands, the DLE-1 enzyme was more sensitive to contaminants than the previous enzyme BspQI. The alternative, in the case of unsuccessful optical map preparation, is to switch to Hi-C technology, for which sequencing libraries and bioinformatic analysis are easier. Indeed, Hi-C can produce chromosome-scale assemblies, even if the anchoring of complex regions, such as centromeres, is less effective, and results should be reviewed with attention.

## 5. Conclusions

The assembly of genomes, and in particular plant genomes, is a challenging field which is undergoing major technological and software evolutions. It generally requires the combination of several technologies, with which it is important to be familiar with, to obtain high-quality results. In this study, we share our experience of reconstructing the genome of *Brassica rapa* (cv. Z1) using long-read sequencing and long-range scaffolding techniques. In our opinion, Bionano Genomics optical mapping is a good choice for organizing and validating long-read assemblies. It anchors the greatest amount of nucleotides, especially in complex regions, such as those with a high proportion of repetitive elements. In addition, it allows estimation of the size of the gaps, unlike scaffolding methods based on Hi-C (or Pore-C) data. However, Hi-C technology is the most popular scaffolding technique, and represents a powerful long-range technology with an active community regularly developing tools and methods. Indeed, the requirement in terms of input material and its sequencing on widely distributed Illumina sequencers makes it an commonly chosen technology. Using only one or carefully combining two methods, we have organized the *Brassica rapa* long-read contigs into a chromosome-scale assembly and obtained the most contiguous genome assembly for this species to date.

## Figures and Tables

**Figure 1 biology-10-00732-f001:**
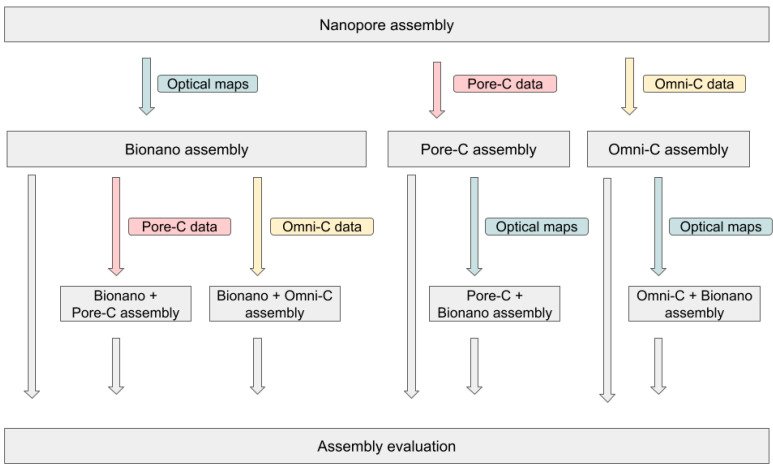
Global view of the different scaffolding experiments. Different scaffolding techniques were applied to the polished Nanopore contigs. Single-technology strategies were first tested. These include Bionano, Omni-C, and Pore-C scaffolding. In a second phase, scaffolds obtained previously were scaffolded again using a different method. In particular, Bionano scaffolds were super-scaffolded using either Pore-C (BNG + Pore-C) or Omni-C (BNG + Omni-C). Pore-C and Omni-C scaffolds were super-scaffolded using Bionano optical maps (respectively, Pore-C + BNG and Omni-C + BNG).

**Figure 2 biology-10-00732-f002:**
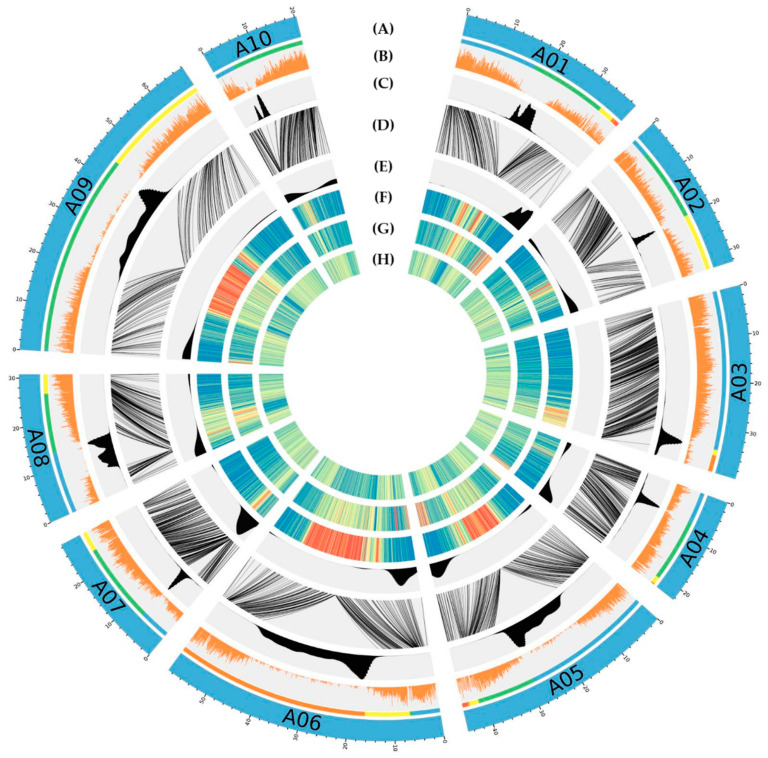
Circular representation of the 10 chromosomes obtained in this novel *B. rapa* cv Z1 assembly V2. (**A**) Scaffolds used to generate each pseudomolecule; (**B**) gene density; (**C**) putative position of (peri)centromeres; (**D**) relationships between the physical and genetic position of the SNP markers used to create the *B. rapa* genetic maps and order the scaffolds onto pseudomolecules; (**E**) density of subtelomeric satellite repeats; (**F**) density of Gypsy elements; (**G**) density of Copia elements; and (**H**) density of DNA retrotransposons.

**Table 1 biology-10-00732-t001:** Metrics of scaffolds generated by using only one scaffolding technology, compared to input contigs. Statistics were generated using sequences of more than 30 kb in size.

	Input Contigs (Necat Assembly)	Bionano	Omni-C	Pore-C
Cumulative size	443,649,441	443,951,349	439,638,897	443,677,941
Number of sequences	299	236	590	253
N50 (L50)	10,461,875 (12)	17,017,634 (8)	25,523,596 (7)	20,151,380 (9)
N90 (L90)	857,267 (58)	3,409,175 (30)	221,999 (98)	1,472,408 (41)
auN	14,202,687	20,478,883	22,995,704	16,823,684
Max. size	45,115,632	44,069,534	42,018,994	32,180,808
Number of Ns (%)	0 (0%)	2,914,945 (0.66%)	20,900 (0.00%)	28,500 (0.01%)
Complete busco genes (%)	1604 (99.4%)	1604 (99.4%)	1604 (99.4%)	1604 (99.4%)
Merqury score	36.4423	37.1176	36.4875	36.4872

**Table 2 biology-10-00732-t002:** Metrics of scaffolds generated by using a combination of different scaffolding technologies, compared to input contigs. Statistics were generated using sequences of more than 30 kb in size.

	Input Contigs (Necat Assembly)	Bionano + Omni-C	Bionano + Pore-C	Omni-C + Bionano	Pore-C + Bionano
Cumulative size	443,649,441	440,038,627	443,961,849	445,844,245	450,760,401
Number of sequences	299	511	222	401	215
N50 (L50)	10,461,875 (12)	33,316,896 (5)	17,017,634 (8)	20,321,816 (7)	25,915,290 (7)
N90 (L90)	857,267 (58)	275,999 (61)	3,409,175 (29)	1,763,661 (32)	3,720,451 (26)
auN	14,202,687	34,864,156	20,976,778	22,582,099	22,825,029
Max. size	45,115,632	64,589,792	44,069,534	51,305,606	43,928,997
Number of Ns (%)	0 (0%)	2,930,045 (0.67%)	2,925,445 (0.66%)	10,143,940 (2.28%)	3,522,960 (0.78%)
Complete buscos genes (%)	1604 (99.4%)	1604 (99.4%)	1604 (99.4%)	1604 (99.4%)	1604 (99.4%)
Merqury score	36.4423	37.1179	37.1176	37.0247	37.0566

**Table 3 biology-10-00732-t003:** Metrics of the final version of *Brassica rapa* cv Z1 Version 2.

	Input Contigs (Necat Assembly)	Bionano	*Brassica rapa* cv Z1 V2	*Brassica rapa* cv Z1 V1 [27]
Cumulative size	443,649,441	443,951,349	443,953,949	401,164,957
Number of sequences	299	236	210	237
N50 (L50)	10,461,875 (12)	17,017,634 (8)	39,217,720 (5)	34,481,996 (5)
N90 (L90)	857,267 (58)	3,409,175 (30)	4,034,065 (13)	2,865,407 (12)
auN	14,202,687	20,478,883	38,695,451	36,201,043
Max. size	45,115,632	44,069,534	68,194,707	57,670,803
Number of Ns (%)	0 (0%)	2,914,945 (0.66%)	2,917,545 (0.66%)	32,966,574 (8.22%)
Number of contigs	299	301	295	297
Contigs N50 (L50)	10,461,875 (12)	10,256,333 (14)	10,256,333 (14)	6,651,009 (14)
Complete busco genes (%)	1604 (99.4%)	1604 (99.4%)	1604 (99.4%)	1594 (98.7%)
Merqury score	36.4423	37.1176	37.119	28.4862
Number of genes	-	-	56,073	46,721
Number of exons/gene	-	-	4.39	4.72
Complete busco genes (%)	-	-	1573 (97.5%)	1553 (96.2%)
Duplicated busco genes (%)	-	-	226 (14.0%)	216 (13.4%)
Fragmented busco genes (%)	-	-	16 (1.0%)	20 (1.2%)
Missing busco genes (%)	-	-	25 (1.5%)	41(2.6%)

## Data Availability

The genome assembly and gene predictions are freely available at http://www.genoscope.cns.fr/plants (accessed on 30 July 2021). The Illumina, and the bionano data of the *B. rapa* cv Z1 version 1 are available in the European Nucleotide Archive under the following projects: PRJEB26620. The PromethION sequencing data, the Omni-C Illumina sequencing data, the Pore-C nanopore sequencing data and the new optical maps are available in the European Nucleotide Archive under the following projects: PRJEB46167.

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
