# Peer review of "Sequencing and Chromosome-Scale Assembly of Plant Genomes, Brassica rapa as a Use Case"

_biology, 2021, doi:10.3390/biology10080732_

Round 1

Reviewer 1 Report

The MS presents a protocol for sequencing and chromosome-scale assembly of plant genomes, using the Brassica rapa as an example. In the lengthy introduction, the authors justified the selection of this plant by being a member of the genus Brassica. This genus is known to include important crops that are cultivated worldwide as vegetables or for oil production. In addition, these species have been regarded as one of the best models to study the importance of polyploidy in plant evolution, diversification, and adaptation, due to the occurrence of both ancient and recent polyploidization events. Too many authors have contributed to the article and the stage seems wide enough to for comparison of Brassica species with different ploidy levels and/or having different basic chromosome numbers.   

However, the introduction seems to be too long and some general information may be deleted.  The materials and methods section is also lengthy and is divided into 13 subsections but the last subsection is also numbered 2.12. (line 299 and line 309).

The discussion could have been made in some more detail and maybe ended with a conclusion statement pointing to the major finding of the results.

Finally, the text is aligned to the left leaving a wide left border blank space. The text should be justified to leave appropriate left and right page borders.

Reviewer 2 Report

As indicated in the Simple Summary, this paper will be extremely useful for those attempting to obtain a high quality and chromosome-organized genome assembly, and more importantly, it describe a technology which makes it afordable for most users. Authors provide information about the state of the art techniques used nowadays. Personally, I found valuable the description of the many programs and pipelines to accomplish this. I understand that authors aim to describe and review the main techniques and approaches to the final user, so many of my own comments go to ensure this goal.  The use, comparison and combination of the three scaffolding approaches gives that extra point of value that will make this paper more relevant and publishable

Some general critiscisms:

  1. Optical maps obtain a linear information of the genome. Other scaffolding approaches such as Omni-C use a different approach that reveal sequences that are not linearly adjacent one to the other that can lead to a different assembly. Authors should discuss more deeply this and reveal the potential pitfalls when reads are scaffolded using a join combination of these techniques

Very Minor points 

  • lane 32. Include "three long-range scaffolding techniques" 
  • lane 75. To clarify to the novel, indicate that HiFi reads are produced by obtaining a consensus after the multiple passes through the same circularized sequences. The description of "multiple passes of the same enzyme" is somehow confusing
  • lane 154. Include the genome size of Brassica. Although this information can be revealed from information provided in lane 332 and others, this will avoid extra calculations to the reader.
  • lane 186. As this paper is intended to teach about these techniques, readers will appreciate the inclusion of a new and short paragraph describing why a nuclease flush is used in the ONT technology. This could be either include here or in the introduction section
  • lane 193: include Wtdbg2/Redbean to not get confussed with the different name used for this same program in lane 197. In lane 335 Wtdbg2 is used again. Be consistent throughout all the document at the time of naming this program
  • Paragraph 2.3: I miss the information about the size and type of Illumina reads used in the polishing of the long-reads. Authors should indicate this. If Illumina reads come from different sources, it should be also indicated. This information will be useful for potential users.
  • Dotplots and the circle figures should be included in the main paper body and not in supplementary data, as they can reveal by the naked eye the quality and the features of the assemblies
  • lane 311, 312 and the whole manuscript:  Write in itallics the scientifical plant names
  • Lanes 342-345: Wondering why the new Necat assembly is the one considered incorrect when no information is provided about the correctness of the Chiifu v3 and the Darmor-bzh reference genomes. If these two latter genomes have been obtained through short reads technology and old-fashioned scaffolding (mate paired and the like), it could be possible that these two are the wrong assembled ?.  Please explain. Maybe in introduction a paragraph explaining how the current Brassica genomes used as (good) reference have been attained that will help in the discussion. For example, lane 482 indicates that Hi-C scaffolding did not generate chimeras when compared with the reference genomes, but this could be due if the previous reference genomes used this scaffolding proccess as well.
  • No discussion about haplotype assembly has been done throughout the document. If authors consider that these genomes were colapsed, they need to make a mention to it.
